

# Assimilation of temperature and relative humidity observations from personal weather stations in AROME-France

Alan Demortier[a], Marc Mandement[a], Vivien Pourret[a], and Olivier Caumont[a,b]

[a]CNRM, Université de Toulouse, Météo-France, CNRS, Toulouse, France
[b]Météo-France, Direction des opérations pour la prévision, Toulouse, France

**Correspondence:** Alan Demortier (alan.demortier@meteo.fr)

**Abstract.** Personal weather station (PWS) networks owned by citizens now provide near-surface observations at a spatial density unattainable with standard weather stations (SWSs) deployed by national meteorological services. This article aims to assess the benefits of assimilating PWS observations of screen-level temperature and relative humidity in the AROME-France model, in the same framework of experiments carried out to assimilate PWS observations of surface pressure in a previous work. Several methods for pre-processing these observations, in addition to the usual data assimilation (DA) screening, are evaluated and selected. After pre-processing, nearly 4700 temperature and 4200 relative humidity PWS observations are assimilated per hour, representing 3 and 6 times more than SWS observations, respectively. Separate assimilation of each variable in the atmosphere with the 3DEnVar DA scheme significantly reduces the root-mean-square deviation between SWS observations and forecasts of the assimilated variable at $2\,\mathrm{m}$ height above ground level up to $3\,\mathrm{h}$ range. Improvements to the near-surface temperature and relative humidity fields analysed are shown for a sea breeze case during a heatwave and a fog episode. However, degradation of short-range forecasts are found when PWS observations are assimilated with the current operational 3DVar DA scheme in the atmosphere or jointly in the atmosphere and at the surface with 3DEnVar and Optimal interpolation DA schemes. These results demonstrate that the benefit of assimilating temperature and relative humidity PWS observations can be highly dependent on the DA schemes and settings employed.

## 1 Introduction

The increase in spatial and temporal resolution of regional numerical weather prediction (NWP) models requires their analyses to be initialized by spatially and temporally dense observations, to represent meteorological phenomena on increasingly finer scales (Gustafsson et al., 2018). At Météo-France, the operational regional NWP system designed to forecast up to $51\,\mathrm{h}$ lead time such phenomena is AROME-France (Seity et al., 2011; Brousseau et al., 2016). AROME-France currently uses a three-dimensional variational (3DVar) DA scheme. This scheme is planned to be replaced by a three-dimensional ensemble variational (3DEnVar) DA scheme, with an up-to-date background error covariance matrix, improving the spread of the information from the observations (Montmerle et al., 2018; Michel and Brousseau, 2021).

Phenomena whose representation in the analyses needs to be improved are meso-$\gamma$ to meso-$\beta$ scale phenomena (2 to $200\,\mathrm{km}$, Orlanski, 1975) such as thunderstorms, breezes or fog (Stull, 1988) which cause substantial thermodynamic changes, particu-



larly in the atmospheric boundary layer (ABL). New observations are emerging to improve analyses in the ABL, from ground-based microwave radiometers (Caumont et al., 2016; Bell et al., 2022; Vural et al., 2024), water vapour lidars (Flamant et al., 2021), or aircrafts (Pourret et al., 2022). Even if these instruments provide dense vertical observations, their horizontal density remains low. Work is underway to exploit horizontally dense satellite observations near the surface, such as radiances from the Meteosat Spinning enhanced visible and infrared imager (SEVIRI) (Sassi et al., 2019), which are currently assimilated with a spacing of the order of 30 km and are still difficult to assimilate at full resolution.

The growth of crowdsourced data, i.e. the data obtained from a range of sensors belonging to the public and shared via the Internet, could play a significant role in improving near-surface analyses of NWP models, in particular data from personal weather stations (PWSs) (Hintz et al., 2019; Coney et al., 2022). In order to obtain observations comparable with standard weather station (SWS) observations, PWS observations should be properly pre-processed, i.e. bias-corrected or quality-controlled or both, the bias designating the deviation of a PWS observation time series from a reference time series (other observations or a modelled field). Indeed, PWSs provide observations that may not comply with World Meteorological Organization (WMO) methods: PWS sensors have heterogeneous environments of siting, sometimes unsuitable shelters, can be positioned at various heights above ground level (AGL), and can suffer from time lag compared to sensors following WMO requirements (Bell et al., 2015; Varentsov et al., 2020; Fenner et al., 2021). After pre-processing, observational studies showed that PWS observations, when combined with SWS observations, are able to describe near-surface thermodynamic variations associated with mesoscale phenomena such as thunderstorms partially visible with SWS observations only (Clark et al., 2018; Mandement and Caumont, 2020).

Various bias correction (BC) methods have been developed. The bias can be computed over a short period (e.g. 6 h for Clark et al., 2018 or approximately 24 h for Mandement and Caumont, 2020). To account for the effects of an inappropriate sheltering or siting, the bias can be computed as a linear function of solar incident radiation, or by a more sophisticated decomposition, e.g. with multilinear functions (Sgoff et al., 2022), generalized additive mixed models (GAMMs) (Cornes et al., 2019), or with machine learning methods (Beele et al., 2022; Marquès, 2023).

Regarding quality control (QC) methods, CrowdQC (Meier et al., 2017) and CrowdQC+ (Fenner et al., 2021) were designed to remove PWS temperature observations deemed erroneous based on their deviation from neighbouring observations. CrowdQC was used in numerous studies of urban temperatures (Feichtinger et al., 2020; Venter et al., 2020; Potgieter et al., 2021; Zumwald et al., 2021). Another QC for PWS temperature observations, used operationally in post-processing algorithms by MET Norway, is called Titan (QC-Titan hereafter, Båserud et al., 2020; Nipen et al., 2020). CrowdQC+ and QC-Titan are comparable in a way as they both use the information from their neighbours to remove inconsistent observations, e.g. the m5 procedure of the CrowdQC+ is comparable to the spatial buddy check of QC-Titan (Fenner et al., 2021). In addition, as CrowdQC+ uses local climate zones, it is challenging to use on a scale larger than a city. For both temperature and relative humidity PWS observations, Mandement and Caumont (2020) have developed a QC (QC-MC hereafter) using an adaptive rejection threshold based on a comparison to interpolated SWS observations. Both QC-Titan and QC-MC are tested in this study.





Once PWS observations have been pre-processed, the opportunity of their assimilation arises. Sgoff et al. (2022) carried

out experiments assimilating pre-processed (bias-corrected and quality-controlled) PWS temperature and relative humidity observations – using the flow-dependent local ensemble transform Kalman filter (LETKF) DA scheme of the Icosahedral non-hydrostatic model with 2 km resolution (ICON-D2). They showed that the assimilation of PWS observations is beneficial; however, no experiments assimilating at the same time both SWS and PWS observations were conducted. On the other hand, Demortier et al. (2024, hereafter D24) carried out experiments assimilating simultaneously pre-processed PWS and

SWS observations of surface pressure with both 3DVar and 3DEnVar DA schemes in AROME-France. Statistically significant improvements of mean sea level pressure forecasts up to 9 h range were found with the 3DEnVar scheme.

This article extends Sgoff et al. (2022) and D24 works by simultaneously assimilating pre-processed PWS and SWS observations of screen-level temperature and relative humidity, and address the following questions. What are the most effective pre-processing methods for assimilating PWS data? Are AROME-France's analyses and forecasts improved by assimilating

pre-processed PWS temperature and relative humidity observations? What impact does the choice of the DA scheme have?

The remainder of this article is organized as follows. Section 2 describes the AROME-France NWP system and its DA schemes in the atmosphere and at the surface, as well as the observations used and the pre-processing methods of these observations. Section 3 describes the assimilation experiments. Objective results of these experiments are given in Sect. 4 and results for case studies are detailed in Sect. 5. Finally, results are summarized and discussed in Sect. 6.

## 2  Datasets and methods


The assimilation experiments run from 6 September to 5 October 2021, as in D24. This one-month study period encompasses a diverse range of meteorological events. It includes two periods of anticyclonic conditions, during which a heatwave (from 2 to 8 September) and fog (from 22 to 24 September) were observed. It also includes disturbed weather episodes such as squall lines on 8 September in the south-west of France, a mesoscale convective system in the south-east of France on 14 September,

and heavy precipitation events from 2 to 4 October.

### 2.1  AROME-France NWP system

AROME-France is the limited area NWP model developed by Météo-France, operational since December 2008 (Seity et al., 2011; Brousseau et al., 2016). AROME-France is a spectral model coupled to the global NWP system ARPEGE (Courtier et al., 1998). AROME-France (AROME hereafter) has a 1.3 km grid spacing on the horizontal and has 90 vertical levels in the

atmosphere ranging from 5 m height AGL (lowest atmospheric model level) up to 10 hPa.

AROME uses a mass-flux shallow convection scheme (Pergaud et al., 2009). Its associated surface scheme is SURFEX (Masson et al., 2013). AROME and SURFEX exchange flux and near-surface variables such as temperature, humidity, and wind. Surface variables are computed over tiles representing four types of surfaces: nature, town, inland water, and ocean. The surface layers have a dedicated assimilation system, which is described at Sect. 2.1.2.



AROME has a 1 h DA cycle for the atmosphere, and a 3 h DA cycle for the surface. From the analysis, the experiments in this study produce 24 h forecasts at 00:00, 06:00, 12:00 and 18:00 UTC.

### 2.1.1 AROME atmospheric DA system

The atmospheric DA procedure is composed of two main steps, which are the screening, which is a series of QC checks, and the minimization of the cost function used for variational DA (see D24). During both steps, modelled variables are compared

to observed variables by an observation operator. AROME has 13 prognostic variables and 5 control variables, which include temperature and specific humidity. Relative humidity is a diagnostic variable computed from specific humidity, temperature, and pressure. The formula used in AROME is given in Appendix A. The model equivalents of observed screen-level temperature (T) and relative humidity (RH) are 2 m height AGL temperature ($T_{2\mathrm{m}}$) and relative humidity ($RH_{2\mathrm{m}}$), described by Vasiljevic et al. (1992) and ECMWF (2023). Computation of $T_{2\mathrm{m}}$ uses the temperature from the lowest atmospheric model

level and the surface temperature, based on the Monin–Obukhov theory – with separated solutions for stable and unstable conditions (Cardinali et al., 1994). $RH_{2\mathrm{m}}$ is computed using only the lowest level fields of the atmospheric model.

The difference between the observation and the model equivalent from the background (1 h forecast starting from the previous analysis), i.e., the observation minus the background, is further called OmB. In the same way, the difference between the observation and its corresponding value in the analysis is further called OmA.

One of the main QC checks of the screening reject observations verifying:

$$\|\mathrm{OmB}\| > \frac{\alpha}{\sigma_{\mathrm{coef}}}\sqrt{\sigma_o^2 + \sigma_b^2} \tag{1}$$

with $\alpha = 4$ and $\sigma_{\mathrm{coef}} = 0.9$, two coefficients; $\sigma_o$ is the standard deviation of observation errors set to 1.4 °C and 10 %, for screen-level T and RH observations; $\sigma_b$ is the standard deviation of background errors. As an example, for $T_{2\mathrm{m}}$, during the first hour of the assimilation experiments (i.e., the 6 September 2021 at 01:00 UTC), the average $\sigma_b$ equals 0.5 °C and the

rejection threshold is up to 6.5 °C. For $RH_{2\mathrm{m}}$, the average $\sigma_b$ equals 10 % and the rejection threshold is up to 70 %. These rejection thresholds are high because current near-surface observations assimilated (SWS observations) are considered as anchor observations for the model, and screening has been designed to keep as much of these observations as possible.

All active observations, i.e. those that have passed the screening, are combined with the background to produce the analysed state of the atmosphere. AROME currently uses a 3DVar DA scheme to constitute the atmospheric analysis fields. The only

difference between the 3DVar and the 3DEnVar DA schemes comes from the background error covariance matrix **B** which is static (prescribed) for the 3DVar and dynamic for the 3DEnVar, estimated at each hour from the AROME Ensemble Data Assimilation system (EDA). To illustrate the difference between the two schemes, the increments (i.e. the analysis minus the background) at the lowest atmospheric model level when single screen-level T and RH observations are assimilated in two idealized experiments are shown in Fig. 1. The propagation of the information from an observation is isotropic with the 3DVar

DA scheme (Fig. 1a and c), whereas with the 3DEnVar DA scheme it is anisotropic and localized within 25 km around the observation (Fig. 1b and d). At the observation location, the values of the increments differ. From an identical OmB of 3.4 °C in both experiments, the increment at the observation location reaches 1 °C with the 3DVar DA scheme, and is reduced to



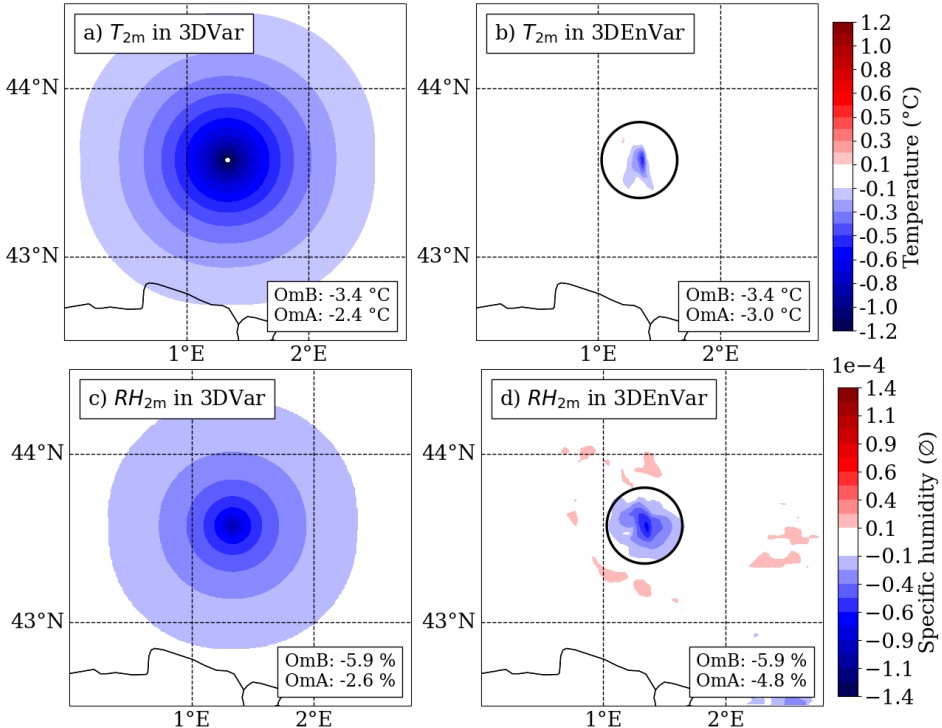

**Figure 1.** Increments at the lowest atmospheric model level of (a, b) temperature and (c, d) specific humidity, resulting from the assimilation of screen-level T and RH observations, respectively, at Blagnac SWS on 6 September 2021 at 01:00 UTC with the (a, c) 3DVar and (b, d) 3DEnVar DA schemes. The black circle indicates a 25 km distance around the observation location.

0.4 °C with the 3DEnVar DA scheme. Similarly, the assimilation of a single $RH_{2m}$ observation results in an increment of 3.3 % (resp. 1.1 %) with the 3DVar (resp. 3DEnVar) DA scheme. The relative observation weight is diminished from the 3DVar to the 3DEnVar DA scheme due to lower $\sigma_b$ on average during the study period for temperature observations. Further details on the 3DVar and the 3DEnVar DA schemes configurations are provided by D24.

### 2.1.2 AROME surface DA system

AROME has a dedicated surface DA scheme for the initialization of the soil prognostic variables, which consist of two ground temperatures and two ground water contents (Giard and Bazile, 2000). The surface DA scheme has its own screening, which uses the same formulation as the atmospheric one (Eq. 1) with, for $T_{2m}$ (resp. $RH_{2m}$), $\alpha$ set to 5 (resp. 2.5), $\sigma_{coef}$ set to 1, $\sigma_o$ set to 1.3 °C (resp. 10 %) and $\sigma_b$ set to 1.6 °C (resp. 10 %).

The surface DA scheme is based on atmospheric increments of $T_{2m}$ and $RH_{2m}$. First, $T_{2m}$ and $RH_{2m}$ are computed with the observation operator described in Sect. 2.1.1. Then, a univariate optimal interpolation (OI) scheme combines screen-level observations with $T_{2m}$ and $RH_{2m}$ background, using isotropic structure functions. The characteristic distance $D$ used in the OI




scheme is set to $100\,\mathrm{km}$ for $T_{2\mathrm{m}}$ and $RH_{2\mathrm{m}}$ observations, which is a compromise between the resolution of the NWP model, the scale of the weather phenomenon represented and the density of the weather stations. Then $T_{2\mathrm{m}}$ and $RH_{2\mathrm{m}}$ increments are used to correct the soil variables with linear interpolations. More details on the surface DA are given by Sassi et al. (2019).

### 2.1.3 A posteriori diagnostics

A number of diagnostic tools exist to understand a posteriori the way in which the DA schemes use the observations. Both the
Desroziers diagnostic of observation error and the spatial Desroziers diagnostic (also used in D24) are used (Desroziers et al., 2005). The spatial Desroziers diagnostic is computed to define a minimum thinning length $\lambda$, from which the correlated observation errors are found to be significantly low. Then, the observation network is thinned by selecting one random observation per mesh from a horizontally-spaced $\lambda$ grid (D24).

## 2.2 Screen-level observations of temperature and relative humidity

A distinction is made here between standard weather stations (SWSs) and personal weather stations (PWSs): the former are bought, operated and maintained by national meteorological and hydrological services (NMHS), while the latter are managed by private individuals or third-party organizations. Behind this distinction, it is important to keep in mind that citizens or third-party organizations can acquire stations identical to NMHSs, just as NMHSs can use low-cost stations dedicated to citizens.

### 2.2.1 Standard weather station (SWS) observations

This study uses only SWS observations assimilated in AROME, which constitute a subset of all SWS observations. These observations originate from three types of reports: manual land SYNOP, automatic land SYNOP and the French RADOME (Tardieu and Leroy, 2003). The number of temperature (resp. relative humidity) observations from SWSs is 2440 (resp. 1600) in average per hour in AROME. Only 65 % of SWS temperature observations entering the AROME DA system are located in France. This is only 48 % for relative humidity. Over France, there are more than twice as many observations of temperature
as there are observations of relative humidity, and both are fairly evenly distributed (Fig. 2). Regarding the height of observation, air temperature and relative humidity should be and are generally measured between 1.25 and $2\,\mathrm{m}$ height AGL (World Meteorological Organization, 2023), the rule being $1.5\,\mathrm{m}$ in France, often a little higher up in mountainous areas with heavy snowfall. For SWS wind speed observations used in Sect. 4, wind speed is generally measured at $10\,\mathrm{m}$ height AGL.

### 2.2.2 Personal weather station (PWS) observations

Three networks of PWSs are used in this study: Netatmo PWSs, StatIC PWSs and Toulouse Métropole PWSs. In contrast to SWS observations, PWS observations are not currently assimilated operationally in AROME.

The Netatmo PWS network is the largest available in near-real time in France (Mandement and Caumont, 2020). As they are owned by citizens, PWSs are unevenly distributed in France (Fig. 2). The outdoor module of the Netatmo PWS contains temperature and relative humidity sensors. With appropriate sheltering, these sensors have a median and a 95 % range of depar-



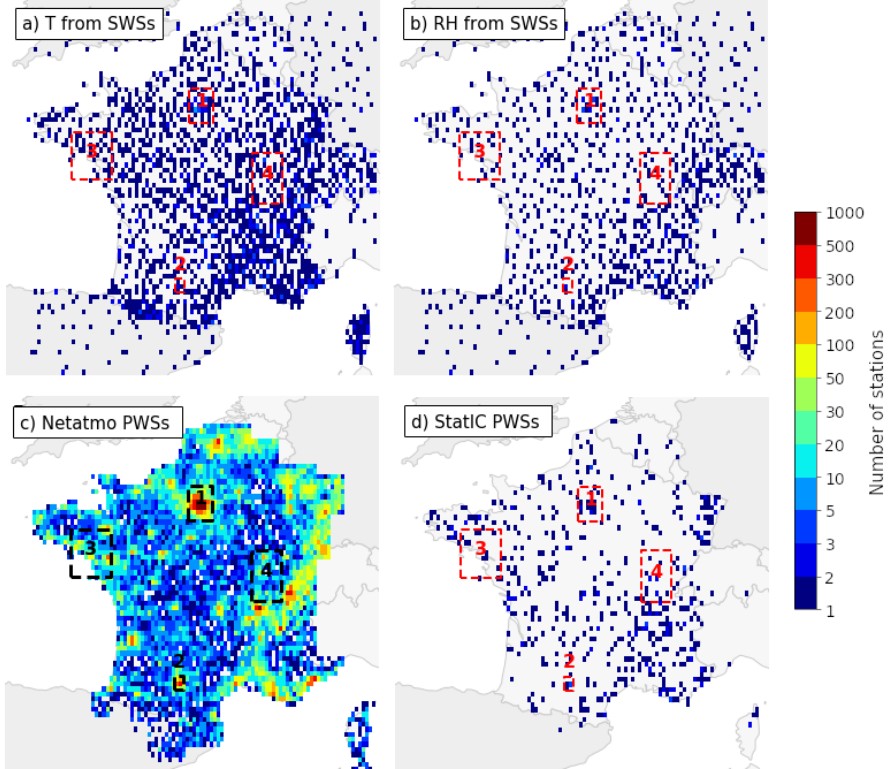

**Figure 2.** Number of (a) SWSs measuring temperature, (b) SWSs measuring relative humidity, (c) Netatmo PWSs and (d) StatIC PWSs providing at least one observation during the one-month study period over France. Observation counts are binned into approximately 0.15 × 0.1° bins. The dashed rectangles delimit the 4 domains used in the cases studied: (1) Paris, (2) Toulouse, (3) along the Atlantic coast for the heatwave case, and (4) the Saône Valley for the fog case.

tures from a reference sensor of about $0\,°\mathrm{C} \pm 0.9\,°\mathrm{C}$ in temperature and $3\% \pm 7\%$ in relative humidity, showing their correct intrinsic quality (Mandement and Caumont, 2020). However, the stations as they are sold are not sheltered according to WMO recommendations: short-wave and long-wave radiation affect $T$ and $RH$, causing departures to sheltered reference sensors (Büchau, 2018). For temperature, these departures are generally overestimations of warm temperatures and underestimations of cold temperatures. These departures are complex depending on the location of the station in relation to its close environment

(e.g. wall, balcony, garden; Varentsov et al., 2020), and its actual height AGL.

     The StatIC PWS network from the Infoclimat association gathers 880 stations in 2023 (Garcelon et al., 2023), of which almost 600 stations are freely downloadable over the period via their open data portal. It gathers Davis Instruments Vantage Pro2 and Vantage Vue PWSs, and also sensors from Dragino or Talkpool companies in shelters, owned by citizens or by the association itself. To join the network, each station should follow the WMO recommendations adapted by Météo-France

(Leroy, 2014), wherever possible: T and RH observation in a normalized shelter, in a clear environment and at $1.5\,\mathrm{m}$ height





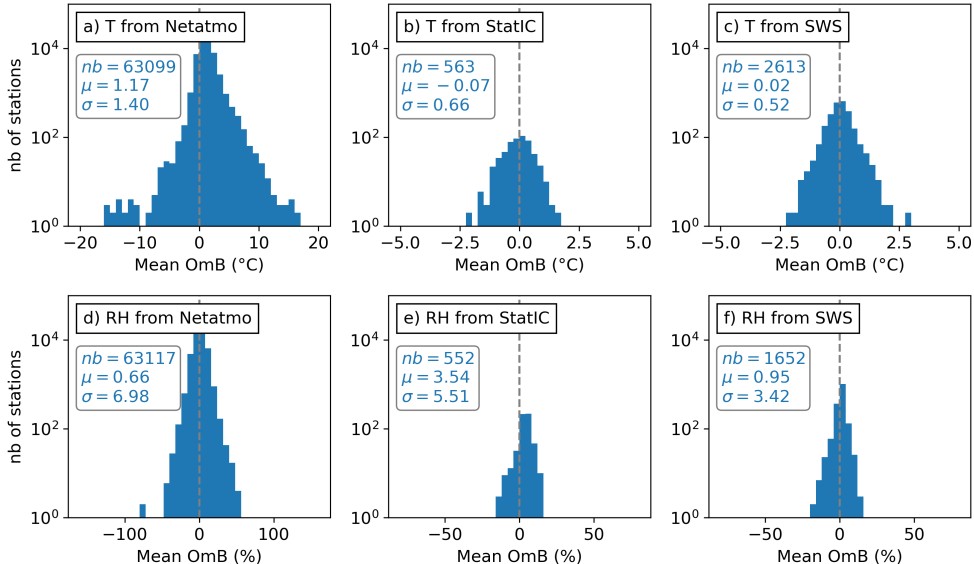

**Figure 3.** Histograms of average OmB over the one-month study period for (a, d) Netatmo PWSs, (b, e) StatIC PWSs, and (c, f) SWSs observations of (a–c) temperature and (d–f) relative humidity. Boxes indicate $nb$ the number of OmB time series, $\mu$ the average OmB for all time series, and $\sigma$ the standard deviation of average OmB for all time series.

AGL. However, in enclosed urban environments (with low air circulation), the recommended height is 1.5 m above the roof, i.e. various heights AGL depending on the height of the roof. StatIC PWS are set up in meteorologically interesting locations such as La Chapelle-en-Vercors which is a cold hole, or areas not well covered by SWS. The accuracy given for the Davis Instruments Vantage Pro2 or Vue PWS is 0.3 °C for temperature and 2 % for relative humidity (Garcelon et al., 2023).

Toulouse Métropole has deployed Davis Vantage Pro2 PWSs around the city of Toulouse for the study of the urban heat island (Dumas et al., 2021). During the study period, 33 PWSs are available.

Figure 3 shows the average OmB distributions for the two largest PWS networks and the SWS network. For temperature over the study period, there are only one fifth as many StatIC PWS observations as SWS observations, while there are about 25 times as many Netatmo PWS observations. However, the heterogeneous siting of the PWSs, seen by the OmB standard

deviation, increases from SWSs to StatIC PWSs to Netatmo PWSs. The results are similar for relative humidity.

In this study, the only PWS observations assimilated are from Netatmo PWS, hereafter referred to as PWS. StatIC and Toulouse Métropole PWS observations are used to evaluate pre-processing methods of PWS observations of temperature. For the evaluation, Netatmo PWS observations are interpolated at the location of StatIC and Toulouse Métropole PWSs, providing an estimate which is compared to the StatIC and Toulouse Métropole PWS observations. The linear interpolation method takes

into account the vertical profile of temperature. The use of independent (i.e. non-assimilated) PWS observations allows to verify the absence of bias in the model.





## 2.3 Satellite observations

Observations of top of the atmosphere bidirectional reflectance from the High resolution visible (HRV) channel from the SEVIRI radiometer, onboard the Meteosat second generation 3 (MSG-3) satellite (also called Meteosat-10) in Rapid Scanning Service (RSS) at $1\,\mathrm{km}$ horizontal resolution every $5\,\mathrm{min}$ are used in Sect. 5.2 to locate clouds. The HRV channel is a broadband channel which is sensitive to wavelengths between 0.4 and $1.1\,\mu\mathrm{m}$ (Schmetz et al., 2002). Reflectance observations from EUMETSAT are corrected depending on the solar angle following the method of Li and Shibata (2006) applied to visible channels in Météo-France operational products (Derrien and Le Gléau, 2010).

An estimate of the global solar radiation at each PWS location has been derived from the spatialized global solar radiation product made by Météo-France in near-real time. This product uses both in situ observations and surface solar irradiance from MSG satellites (eumetsat, 2017). Nighttime values are set to zero. The hourly global solar irradiance is cumulated over each hour and is used as a predictor for one of the bias correction methods of PWS observations.

## 2.4 Pre-processing methods

In the article, PWS observations used are interpolated to round hours and PWSs with identical coordinates are removed, as in the step preparation of D24. Hereafter, these observations, prepared but not pre-processed, are referred to as raw PWS observations.

### 2.4.1 Bias correction (BC) methods

The bias correction (BC) ensure that possible biases (i.e. a systematic departure to a reference) in PWS observation time series do not propagate to the model. In this study, raw PWS observations of screen-level temperature and relative humidity have been monitored, i.e. $T_{2\mathrm{m}}$ and $RH_{2\mathrm{m}}$ OmB time series are computed, in the operational AROME 3DVar configuration, which is called the Monitor experiment (as in Sgoff et al., 2022 and D24). Four types of BC methods that can be applied in real-time, because they use past OmB statistics, are tested. They are named as follows:

- BC-D removes the average OmB of the last $24\,\mathrm{h}$ (average of 24 hourly OmBs, D for daily), in a manner close to Mandement and Caumont (2020).

- BC-M removes the average OmB of the last $30\,\mathrm{d}$ (M for monthly).

- BC-S uses the BC method developed by Sgoff et al. (2022); S for Sgoff. The bias is computed from five predictors composed of trigonometric basis functions from the hour of the measured observations with $2\,\mathrm{d}$ inertia.

- BC-R uses a random forest method trained with the hour of the day and cumulative hourly global solar irradiance predictors from the last $30\,\mathrm{d}$, close to the BC used by Beele et al. (2022), but with fewer predictors (R for radiation).





### 2.4.2 Quality control (QC) methods

Quality control (QC) methods are used in this study to remove PWS observations judged erroneous, because the current screening for near-surface observations has not been designed to deal with observations with OmB statistics so different from those of SWS (Sect. 2.2.2).

Titan or titanlib (QC-Titan, Båserud et al., 2020), is an open-source library composed of various checks or tests. In this article, as in Nipen et al. (2020), Titan-QC designates the combination of three main spatial tests which are the buddy check, the spatial consistency test (SCT), and the isolation test. The buddy check consists in removing observations if their deviation from the average is more than twice the standard deviation of the observations in the neighbourhood within a 15 km radius. The SCT consists in an iterative cross-validation procedure. For each observation, the SCT adjusts a vertical profile, and computes an estimation of a true observation, and the standard deviation of the observations in the neighbourhood. The observation is removed if the ratio of the squared deviation from the observation estimation divided by the standard deviation is greater than 4 (or 8 for negative values). The process is repeated until no observations are removed. The isolation test removes observations which have less than 5 stations within a 15 km radius and 200 m elevation difference.

QC-MC (Mandement and Caumont, 2020) has been adapted for use on a larger scale, throughout France, even though it was designed for the scale of a French region: computations are done simultaneously for 11 selected climatological areas. The second main adaptation is the use, for each PWS observation, of OmB statistics from AROME instead of comparisons between this PWS observation and an estimate derived from neighbouring SWS observations. In QC-MC, every hour, in every climatological area, for each PWS, the rms of PWS OmBs are computed on the last 6 h, to separate observations that differ from the background at a given time due to physical reasons (e.g. a phenomenon not well positioned in the background) to the observations that differ all the time from the background. PWS having rms OmB exceeding an adaptive threshold, computed according to Mandement and Caumont (2020), are removed.

## 3 Assimilation experiments design

### 3.1 Choice of bias correction and quality control

In the literature, some authors carry out a QC without a BC (Nipen et al., 2020). To evaluate this choice, we compare QC-MC and QC-Titan. Note, however, that QC-MC was designed to be used after a BC. Every hour, the observations considered incorrect by the two QCs are removed. The hourly mean OmB for the PWS temperature observations are shown in Fig. 4a. The average OmB exhibits a systematic bias following a diurnal cycle: it varies from about 0.7 °C during the day to 1.3 °C during the night. The systematic warm bias could be due to site effect as the PWSs are placed close to heating environments such as windows or balconies (Bell et al., 2015; Sgoff et al., 2022). For relative humidity (Fig. 5b), the raw PWS observations exhibit a diurnal bias composed of a drier nighttime and a wetter daytime. Its variations are close to what was reported by Sgoff et al. (2022). Regarding the observations after the two QC, the mean OmB is reduced throughout the day when using QC-MC, while it still shows a diurnal bias with QC-Titan (Fig. 4a). QC-MC takes into account a reference, which is here the

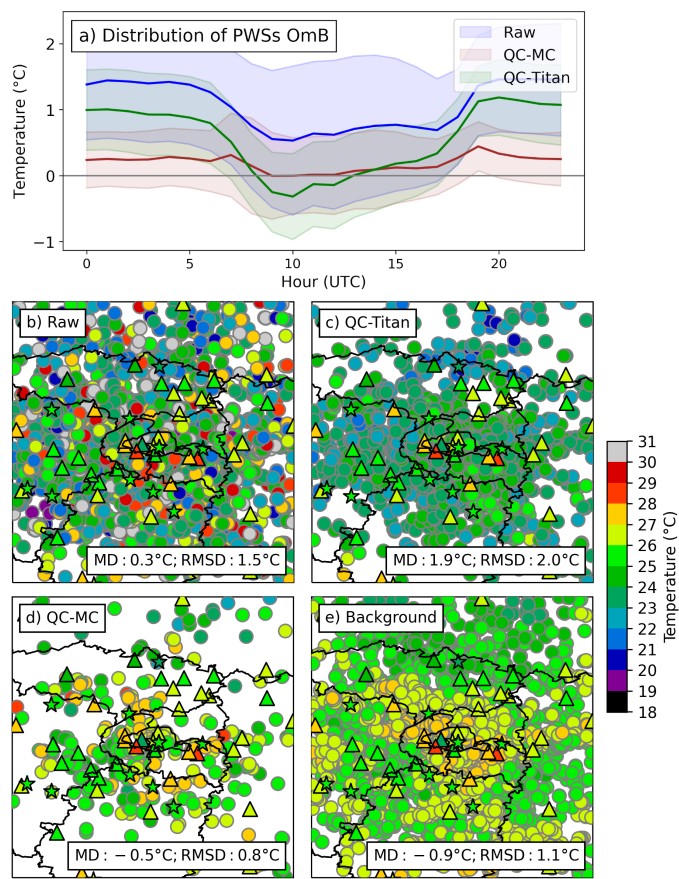

**Figure 4.** (a) Distributions of PWSs OmB for each hour of the study period: thick lines indicate the mean, thin lines indicate the mean $\pm$ one standard deviation. The area between thick and thin lines is shaded. (b–e) Screen-level temperature observations around Paris on 7 September 2021 at 10:00 UTC from (coloured triangles with black contours) SWSs, (coloured stars) StatIC PWSs. Coloured circles are either (b–d) PWS screen-level temperature observations or (e) $T_{2m}$ AROME background at PWS location. PWS observations are (b) raw or quality-controlled by (c) QC-Titan, (d) QC-MC. Boxes indicate the mean deviation (MD) and the rms deviation (RMSD) of PWS observations compared with StatIC PWS independent observations.




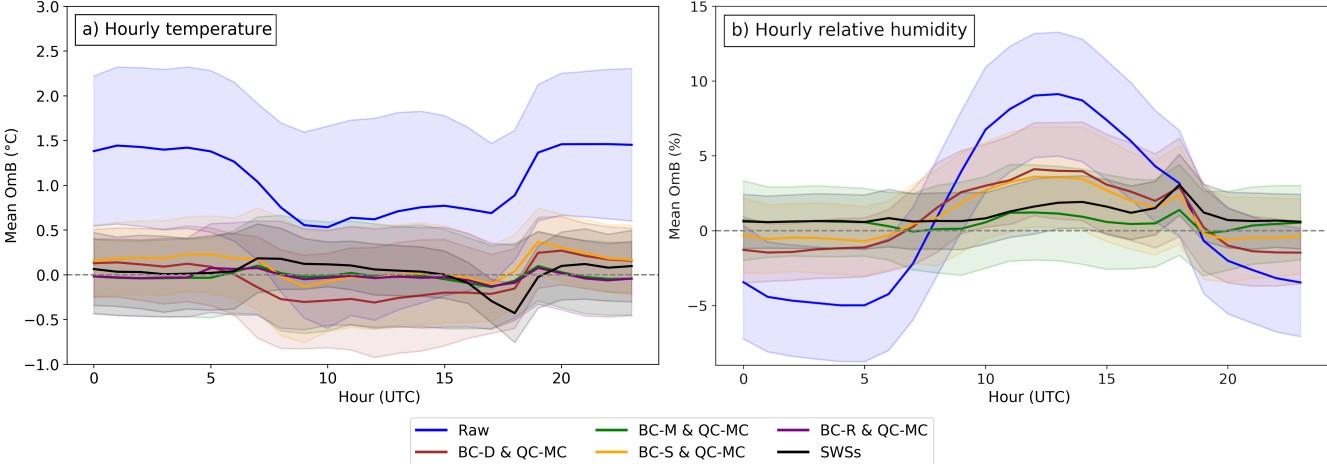

**Figure 5.** Mean diurnal cycle of OmBs during the 1-month study period of (a) temperature and (b) relative humidity for (blue) raw PWS observations, for PWS observations quality-controlled with QC-MC and bias-corrected with (brown) BC-D, (green) BC-M, (orange) BC-S, (purple, only in a) BC-R, and for (black) SWS observations. Lines and colour shades are as in Fig. 4a.

model background, allowing the QC to remove spatially incoherent observation in relation to the model background. A map of quality-controlled temperature observations near Paris at 12:00 LT is shown in Fig. 4b–e. Without QC, no regional pattern could be seen on the temperature observations from PWSs, explaining a high rms deviation to StatIC PWS observations of 1.5 °C

(Fig. 4b). QC-Titan has the ability to remove observations which are too far from their neighbours. After QC-Titan, every single PWS observation is closer to its neighbours, but PWS observations have a mean deviation to StatIC PWS observations of 2.0 °C. When using QC-MC, more stations are removed, but only the observations close to the background are kept (QC-MC PWS OmB threshold is equal to ± 1 °C at the time of the figure). QC-MC will therefore be chosen for the next part of the study. However, QC-MC does not ensure by itself a zero bias of PWS observations: Fig. 4a shows a slight positive bias, and a BC is

necessary to remove it.

To remove the remaining bias, the 4 BC methods described in Sect. 2.4.1 are tested, in association with the selected QC-MC. Figure 5 shows the diurnal cycle of mean OmBs for both the PWS observations after applying the 4 BC methods, and the SWS observations. For temperature observations (Fig. 5a), mean OmBs for all choices of BC are close to zero and are in the order of magnitude of SWS observations OmBs. By construction, BC-M mean OmBs are near zero throughout the diurnal cycle;

BC-R mean OmBs are very close: the addition of hourly global solar irradiance as a predictor does not seem to make any substantial improvement. For relative humidity observations (Fig. 5b), all BCs tested exhibit a reduced diurnal cycle compared to raw PWS observations. BC-D and BC-S mean OmBs are close, as they both use 24 or 48 h rolling periods to estimate biases, respectively. Still, BC-S OmBs are slightly closer to zero during the night. Once again, the BC-M mean OmBs are near 0 % throughout the diurnal cycle, as SWS OmBs.





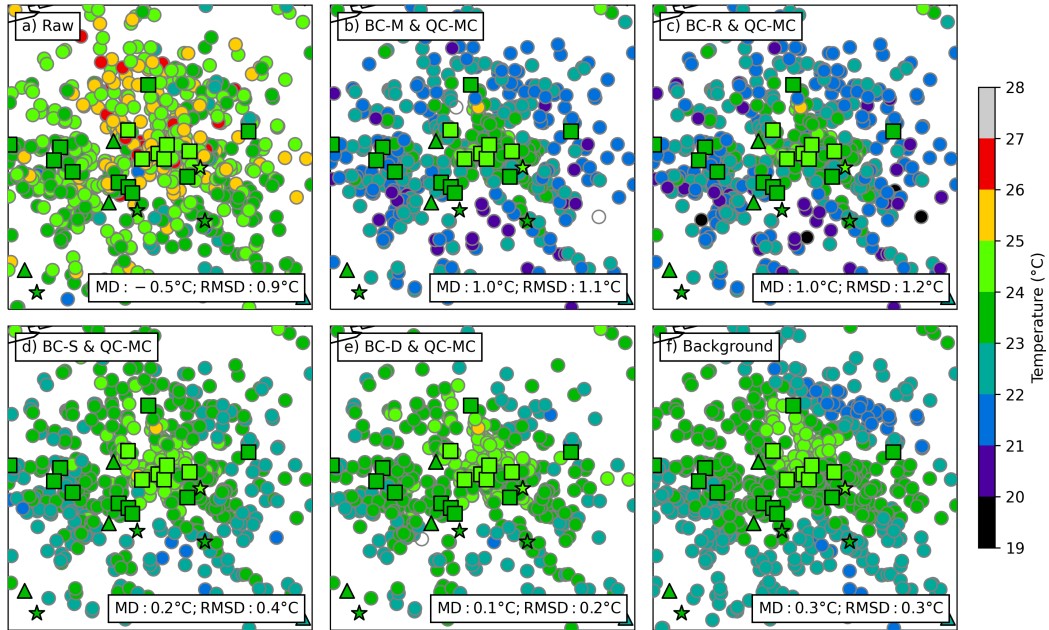

**Figure 6.** As in Fig. 4b–e, around the city of Toulouse on 7 September 2021 21:00 UTC (23:00 LT). Coloured circles are either (a–e) PWS screen-level temperature observations or (f) $T_{2\mathrm{m}}$ AROME background at PWS location. PWS observations are (a) raw, quality-controlled with QC-MC and bias-corrected with (b) BC-M, (c) BC-R, (d) BC-S, (e) BC-D. Coloured squares are Toulouse Métropole PWS screen-level temperature observations. Boxes indicate the mean deviation (MD) and the rms deviation (RMSD) of PWS observations compared with Toulouse Métropole PWS independent observations.

Figure 6 shows how the 4 BC methods modify PWS observations of screen-level temperature on 7 September 2021 23:00 LT around the city of Toulouse (centre of the map). Raw PWS temperature observations are close to observations of SWSs, Toulouse Métropole PWSs or StatIC PWSs networks, even if some PWS observations are warmer north-west of the city. However, BC-M and BC-R which uses monthly rolling periods to estimate biases reduce PWS temperatures in the suburbs of the city, which is inconsistent with other observation networks. In contrast, BC-S and BC-D, which use 24 or 48 h rolling

periods to estimate biases, keep observations in agreement with other observations networks: the RMSD of PWS observations compared with Toulouse Métropole PWS independent observations reaches only 0.4 °C and 0.2 °C, respectively. $T_{2\mathrm{m}}$ of the AROME background (Fig. 6f) are close to Toulouse Métropole observations (RMSD of 0.3 °C), while unfortunately none are available north-east of the city to support the colder temperatures indicated by the background.

## 3.2   Choice of observation error covariances and thinning

In variational DA schemes, the observation error covariance matrix **R** has to be specified. As in D24, the part of **R** allocated to PWS observations is prescribed diagonal, as for the part allocated to SWS observations, and the diagonal values ($\sigma_o$) are the same as for SWS observations: 1.4 °C for $T_{2\mathrm{m}}$ and 10 % for $RH_{2\mathrm{m}}$. This is done for simplicity's sake, as modelling the



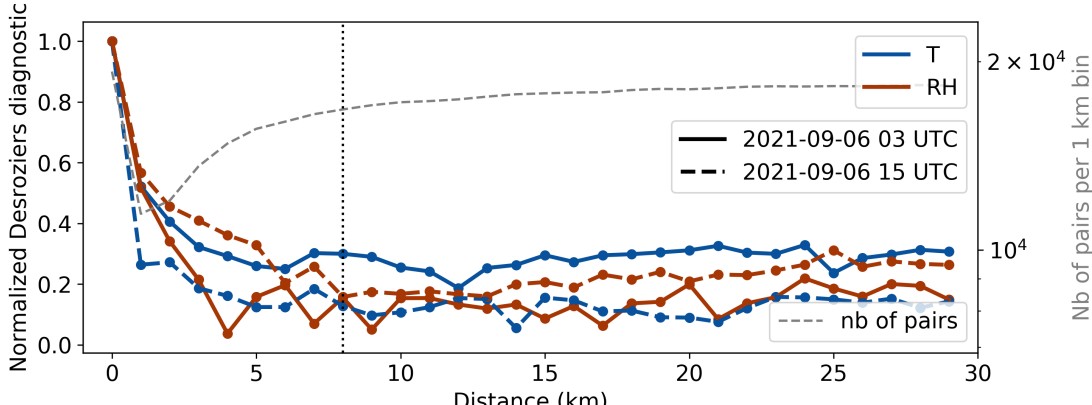

**Figure 7.** Spatial Desroziers diagnostic of PWS observations of (blue) temperature and (red) relative humidity after BC-M and QC-MC on 6 September 2021 at (plain lines) 03:00 UTC and (dashed lines) 15:00 UTC. In grey, the number of pairs of equidistant stations (in 1 km bins).

non-diagonal terms is an open research question (Guillet et al., 2019), and a study of sensitivity to a change in $\sigma_o$ is beyond the scope of this article. The diagonal assumption demands to verify that observations errors are not correlated, and if they are, the
reduction of the spatial density of the observations, may diminish the correlation between observation errors.

Figure 7 shows the spatial Desroziers diagnostic for PWS observations after BC-M and QC-MC. This diagnostic is computed using OmA and OmB of these PWS observations assimilated without thinning in a cycled 3DEnVar DA experiment launched on 6 September 2021 00:00 UTC. This diagnostic decreases when the distance between the observations increases. Two observations being far from each other have less probability of having correlated observation errors, reducing the influence
of their local environment. The spatial Desroziers diagnostic decreases up to distances of 1 to 6 km, depending on the variable and the hour of the day. To reduce the probability of having correlated observation errors, a thinning is applied, selecting one random observation per mesh in an approximately 8 km horizontally spaced regular Gaussian grid (8 km thinning hereafter). One of the limit of the method is the fact that two observations may have correlated errors independently of the distance, for example if they are subject to similar anomalous siting conditions, and these observations are not corrected nor removed by the
pre-processing.

Two pre-processing methods of PWS observations are selected for the DA experiments: P-M composed of BC-M, QC-MC and a 8 km thinning, and P-S composed of BC-S, QC-MC and a 8 km thinning. One of the points to bear in mind when using pre-processing methods based on comparisons with the background of the model into which we want to assimilate these observations is that pre-processed PWS observations can overfit the model background (see the triple collocation method
shown in Appendix B).



**Table 1.** Overview of the experiments. The Monitor experiment is launched on 6 August 2021 00:00 UTC. All cycled experiments are launched on 6 September 2021 00:00 UTC. In the name of the experiments, *X* corresponds to the variable of PWS observations assimilated: T for $T_{2\mathrm{m}}$ or RH for $RH_{2\mathrm{m}}$.

| Experiments | Duration | Cycling | Atmospheric DA scheme | Use of PWS observations |
|---|---|---|---|---|
| Monitor | 2 months | No | 3DVar | Raw observations monitored |
| 3DVar | 1 month | Yes | 3DVar | No |
| 3DVar*X* | 1 month | Yes | 3DVar | Variable *X* assimilated after P-M (BC-M, QC-MC, 8 km thinning) |
| 3DEnVar | 1 month | Yes | 3DEnVar | No |
| 3DEnVar*X* | 1 month | Yes | 3DEnVar | Variable *X* assimilated after P-M (BC-M, QC-MC, 8 km thinning) |
| 3DEnVar*X*S | 1 month | Yes | 3DEnVar | Variable *X* assimilated after P-S (BC-S, QC-MC, 8 km thinning) |
| 3DEnVar*X*-surf | 1 month | Yes | 3DEnVar | Variable *X* assimilated after P-M (BC-M, QC-MC, 8 km thinning) in both atmospheric (3DEnVar) and surface (OI) DA schemes |

## 3.3 Overview of the experiments

All experiments are described in Table 1. The benefit from assimilating PWS observations of a variable *X* (T for $T_{2\mathrm{m}}$ or RH for $RH_{2\mathrm{m}}$) is evaluated using the observing system experiment (OSE) framework, comparing a reference experiment (3DVar, 3DEnVar) to an experiment where PWS observations are assimilated (Pourret et al., 2022; D24). To explore the role of the pre-processing (or bias correction, since this is the only part that changes), PWS observations after P-M and P-S are assimilated using the 3DEnVar DA scheme (3DEnVar*X* and 3DEnVar*X*S, respectively). To explore the role of the DA scheme, PWS observations after P-M are assimilated using the 3DVar DA scheme (3DVar*X*). Finally, the impact of assimilating simultaneously PWS observations after P-M at the surface using the OI DA scheme and in the atmosphere with the 3DEnVar DA scheme is tested (3DEnVar*X*-surf). Monitor, 3DVar and 3DEnVar experiments are the same as in D24. Except Monitor, they are cycled, which means that the assimilation of PWS observations at time $t$ influences the background at time $t+1\,\mathrm{h}$, used to make the new analysis. During one month, 719 analyses are made, and 119 forecasts are launched.

## 4 Results of the assimilation experiments

The experiments are evaluated using statistics of one-month OmB and OmF (observation minus forecast) where observations are systematically SWS observations. The relative evolution of the rms OmX (X being B or F) of an experiment (XP) w.r.t. another (CTRL) is given by:

$$\Delta\mathrm{rms}\,\mathrm{OmX} = \frac{\mathrm{rms}(\mathrm{OmX_{XP}}) - \mathrm{rms}(\mathrm{OmX_{CTRL}})}{\mathrm{rms}(\mathrm{OmX_{CTRL}})} \tag{2}$$

As in D24, $\Delta\mathrm{rms}\,\mathrm{OmX}$ is considered significant if 0 is not in the 95 % confidence interval around it. This interval is computed by bootstrap with the "percentile" method (scipy.stats.bootstrap function, Virtanen et al., 2020): $\mathrm{OmX_{XP}}$ time series is





**Table 2.** $\Delta$rms OmB (%) of surface pressure, $T_{2\mathrm{m}}$, $RH_{2\mathrm{m}}$ and 10 m zonal and meridional wind for SWSs over the one-month study period. Negative values of an experiment (XP) w.r.t. another (CTRL) indicate that the backgrounds (1 h forecasts) of XP are closer than CTRL to SWS observations, i.e. improvement and positive values indicate degradation. Significant values are in bold.

| | Surface pressure | | $T_{2\mathrm{m}}$ | | $RH_{2\mathrm{m}}$ | | Zonal wind | | Meridional wind | |
| --- | --- | --- | --- | --- | --- | --- | --- | --- | --- | --- |
| | AROME domain | France | AROME domain | France | AROME domain | France | AROME domain | France | AROME domain | France |
| 3DVarT w.r.t. 3DVar | 0.2 | 0.7 | **1.8** | **2.4** | −0.1 | −0.3 | −0.3 | −0.5 | −0.1 | −0.2 |
| 3DEnVarT w.r.t. 3DEnVar | 0.1 | **1.6** | **−0.7** | **−0.9** | −0.2 | −0.2 | −0.1 | −0.2 | 0.0 | 0.0 |
| 3DEnVarTS w.r.t. 3DEnVar | **1.3** | **3.7** | **−0.6** | **−0.8** | −0.1 | −0.2 | 0.1 | 0.1 | 0.0 | 0.0 |
| 3DEnVarT-surf w.r.t. 3DEnVar | 0.1 | **0.9** | **0.5** | **0.7** | 0.3 | 0.5 | 0.1 | 0.4 | 0.0 | 0.1 |
| 3DVarRH w.r.t. 3DVar | −0.3 | −0.8 | **−0.6** | **−0.9** | **2.7** | **4.5** | −0.2 | −0.2 | −0.1 | −0.2 |
| 3DEnVarRH w.r.t. 3DEnVar | 0.2 | 0.3 | 0.2 | 0.3 | **−1.0** | **−1.5** | 0.0 | 0.0 | 0.1 | 0.2 |
| 3DEnVarRHS w.r.t. 3DEnVar | 0.5 | 0.7 | **0.5** | **0.8** | **−0.8** | **−1.2** | 0.1 | 0.2 | 0.0 | 0.2 |
| 3DEnVarRH-surf w.r.t. 3DEnVar | 0.1 | 0.1 | 0.3 | **0.4** | **−0.9** | **−1.3** | 0.1 | 0.2 | 0.1 | 0.2 |

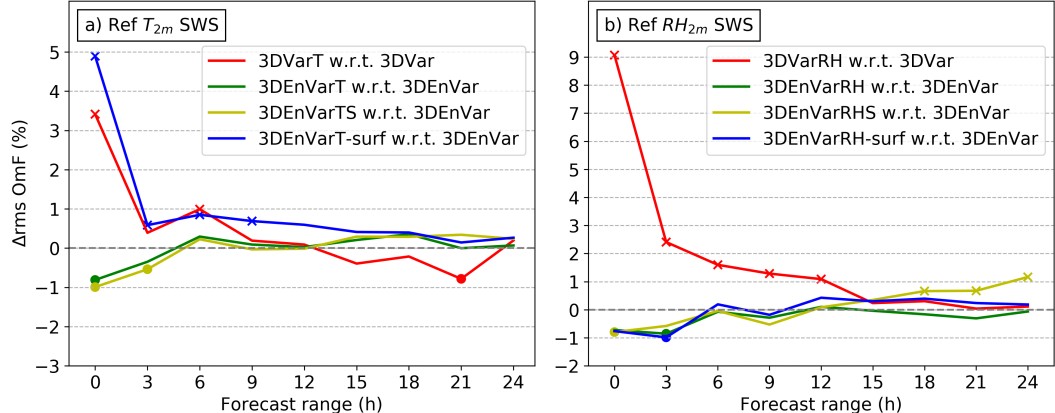

**Figure 8.** $\Delta$rms OmF of (a) $T_{2\mathrm{m}}$ and (b) $RH_{2\mathrm{m}}$ for SWSs over France every 3 h between 0 and 24 h forecast range over the one-month study period. Negative values indicate improvement and positive values indicate degradation. Points (respectively crosses) indicate improvement (resp. degradation) of XP w.r.t. CTRL at 90 % statistical significance level.

randomly sampled with replacement, giving a new time series of the same size. This is done 1000 times, providing 1000 time
series, giving 1000 rms values forming a distribution, from which the confidence interval is estimated.





## 4.1 Impact of the bias correction with the 3DEnVar DA scheme

When PWS $T_{2\mathrm{m}}$ observations are assimilated in 3DEnVarT or 3DEnVarTS (Table 2), $T_{2\mathrm{m}}$ rms OmBs decrease significantly for both experiments in comparison with 3DEnVar, which is an improvement. The improvement is slightly larger with 3DEnVarT ($\Delta$rms OmB of $-0.7\,\%$) than with 3DEnVarTS ($-0.6\,\%$) over the AROME domain. The improvement is larger ($-0.9\,\%$

and $-0.8\,\%$, respectively) over France, as it is the area where PWS observations are assimilated. Surface pressure rms OmB significantly increases over France for 3DEnVarT ($+1.6\,\%$) and 3DEnVarTS ($+3.7\,\%$). Other variables show no significant evolution.

For PWS $RH_{2\mathrm{m}}$ observation assimilation in 3DEnVarRH or 3DEnVarRHS, $RH_{2\mathrm{m}}$ rms OmBs also decrease significantly over the AROME domain ($\Delta$rms OmB of $-1.0\,\%$ and $-0.8\,\%$, respectively). The improvement is larger over France ($-1.5\,\%$

and $-1.2\,\%$, respectively). However, a slight but significant degradation of $T_{2\mathrm{m}}$ rms OmBs is found in the 3DEnVarRHS experiment both over France and the AROME domain ($0.8\,\%$ and $0.5\,\%$, respectively). Other variables show no significant evolution.

For experiments assimilating PWS observations either of $T_{2\mathrm{m}}$ or $RH_{2\mathrm{m}}$, the lowest rms OmBs are obtained with the P-M pre-processing for all variables over France.

Scores for forecasts up to 24 h are shown in Fig. 8. When PWS $T_{2\mathrm{m}}$ observations are assimilated (3DEnVarT and 3DEnVarTS in Fig. 8a), an improvement of $T_{2\mathrm{m}}$ rms OmFs is found up to 3 h forecast range; beyond 3 h forecast range, the rms OmF evolution are neutral to slightly degraded. Similar results are found in 3DEnVarRH and 3DEnVarRHS when PWS $RH_{2\mathrm{m}}$ observations are assimilated. These results are consistent with Sgoff et al. (2022) results showing improvements vanishing or not significant beyond 5 to 6 h forecast range when assimilating bias-corrected PWS observations only.

No significant evolution of rms OmBs or rms OmFs of other observing systems is noticed (not shown), which could be explained by the small impact of $T_{2\mathrm{m}}$ and $RH_{2\mathrm{m}}$ above the ABL (Brousseau et al., 2014). Also, for observing systems such as radiosoundings, the low number of observations over France makes it difficult to show significance (D24).

## 4.2 Impact of the DA scheme

When compared to 3DVar, 3DVarT significantly degrades by $+2.4\,\%$ over France the $T_{2\mathrm{m}}$ rms OmBs (Table 2). For 3DVarRH

compared to 3DVar, this degradation reaches $+4.5\,\%$ . The degradation is very large at the analysis and remains significant between 6 to 12 h forecast range for $T_{2\mathrm{m}}$ and $RH_{2\mathrm{m}}$, respectively (Fig. 8). As it was found by D24 for PWS surface pressure observation assimilation, the 3DVar DA scheme with its operational settings is not able to take advantage of these PWS observations with the pre-processing (including the thinning) which is selected.

When PWS $T_{2\mathrm{m}}$ observations are concomitantly assimilated by the atmospheric and surface DA systems (3DEnVarT-surf),

the improvement found in 3DEnVarT turns off to a significant degradation for $T_{2\mathrm{m}}$ rms OmBs (Table 2). The degradation is significant up to 9 h forecast range (Fig. 8). When PWS $RH_{2\mathrm{m}}$ observations are concomitantly assimilated by the atmospheric and the surface DA system (3DEnVarRH-surf), $RH_{2\mathrm{m}}$ rms OmBs and OmFs are very similar to 3DEnVarRH, showing an improvement. These large differences between the assimilation of PWS observations of $T_{2\mathrm{m}}$ and $RH_{2\mathrm{m}}$ in the surface DA



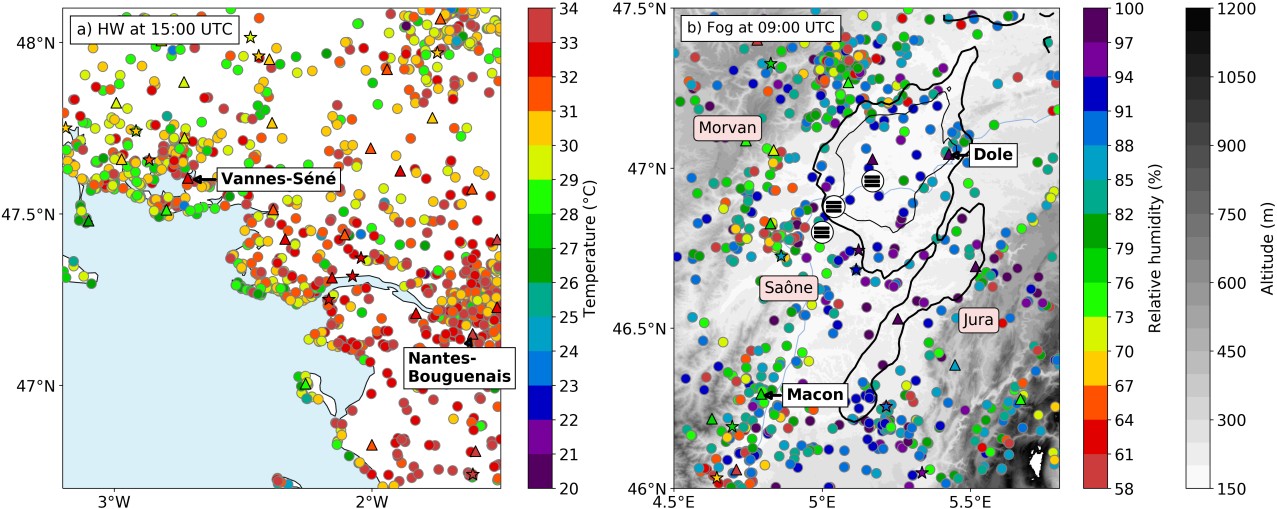

**Figure 9.** Geographical situation of (a) the heatwave case on 7 September 2021 15:00 UTC with raw screen-level temperature observations and (b) the fog case on 23 September 2021 09:00 UTC with raw screen-level relative humidity observations. SWSs are indicated by coloured triangles with black contours, PWSs by coloured circles, and StatIC PWSs by coloured stars. Human crowdsourced observations of fog between 08:00 and 09:00 UTC are indicated by three encircled horizontal bold lines (fog symbol). MSG-3 satellite HRV channel reflectance observations equal to 24 % and 43 % are shown by bold and thin black edges, respectively. Light blue indicates water surfaces (here mainly the Atlantic Ocean).

scheme could be explained by the fact that the $RH_{2m}$ observation operator, contrary to the $T_{2m}$ observation operator, uses only
variables from the lowest atmospheric model level, and not surface variables.

## 5 Results on case studies

In case of a strong near-surface gradient of a meteorological variable caused by a meso-$\beta$ scale meteorological phenomenon, the added value of a dense network of weather stations is likely to be clearly visible. Two cases with substantial temperature and relative humidity gradients in September 2021 are studied: a sea breeze case during a heatwave on 7 September with around
4 °C of screen-level temperature difference between two SWS observations 12 km apart, and a fog case on 23 September with 40 % screen-level relative humidity differences between two SWS observations 25 km apart.

### 5.1 Sea breeze front during a heatwave on 7 September 2021

On 7 September 2021, at the synoptic scale, subtropical warm air came from North Africa, ahead of a low pressure system located over the Atlantic Ocean. Near the surface, the south-eastern to eastern winds carried the warm continental air mass
over western France. Maximum daily temperature reached 33.7 °C at Nantes-Bouguenais SWS at 15:07 UTC and 32.2 °C at 15:00 UTC at Vannes-Séné SWS (Fig. 9a). It is 11.3 and 10.3 °C more than the 1991–2020 September normal maximum

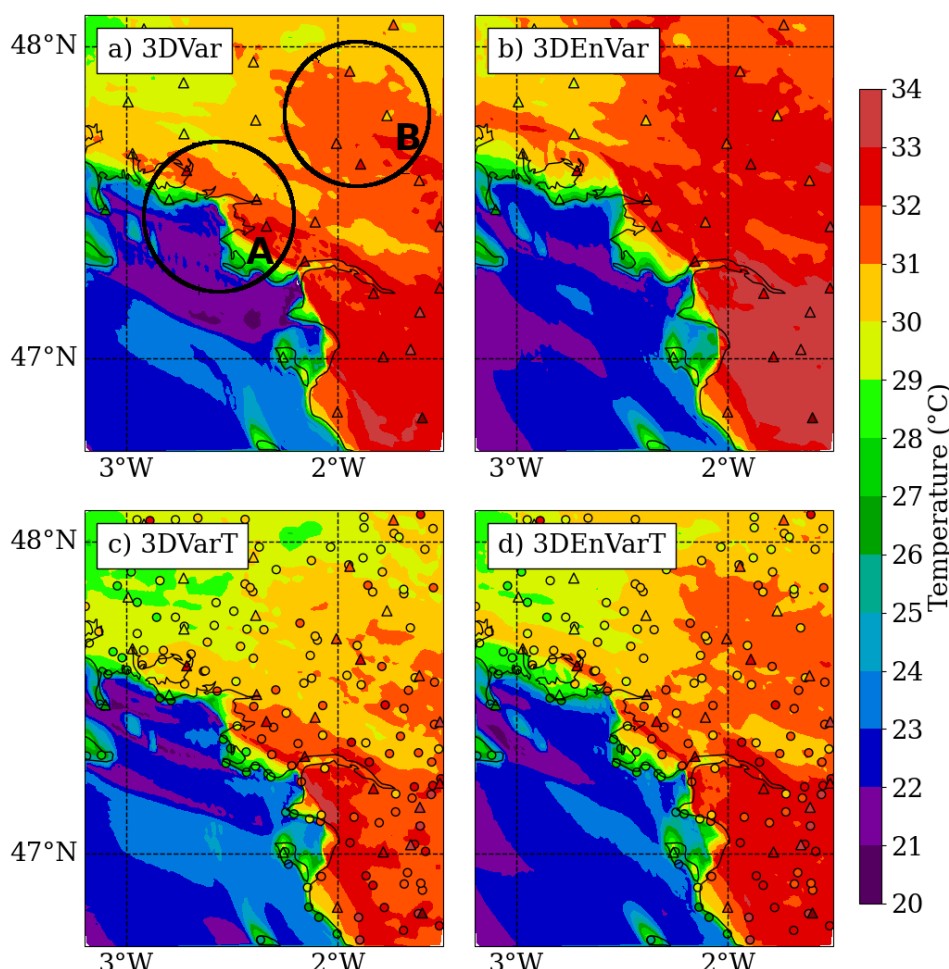

**Figure 10.** Temperature of lowest atmospheric model level (around 5 m height AGL) analyses for (a) 3DVar, (b) 3DEnVar, (c) 3DVarT, and (d) 3DEnVarT experiments on 7 September 2021 at 15:00 UTC. Locations of the stations whose observations of screen-level temperature are assimilated are indicated by triangles for SWSs and circles for PWSs.



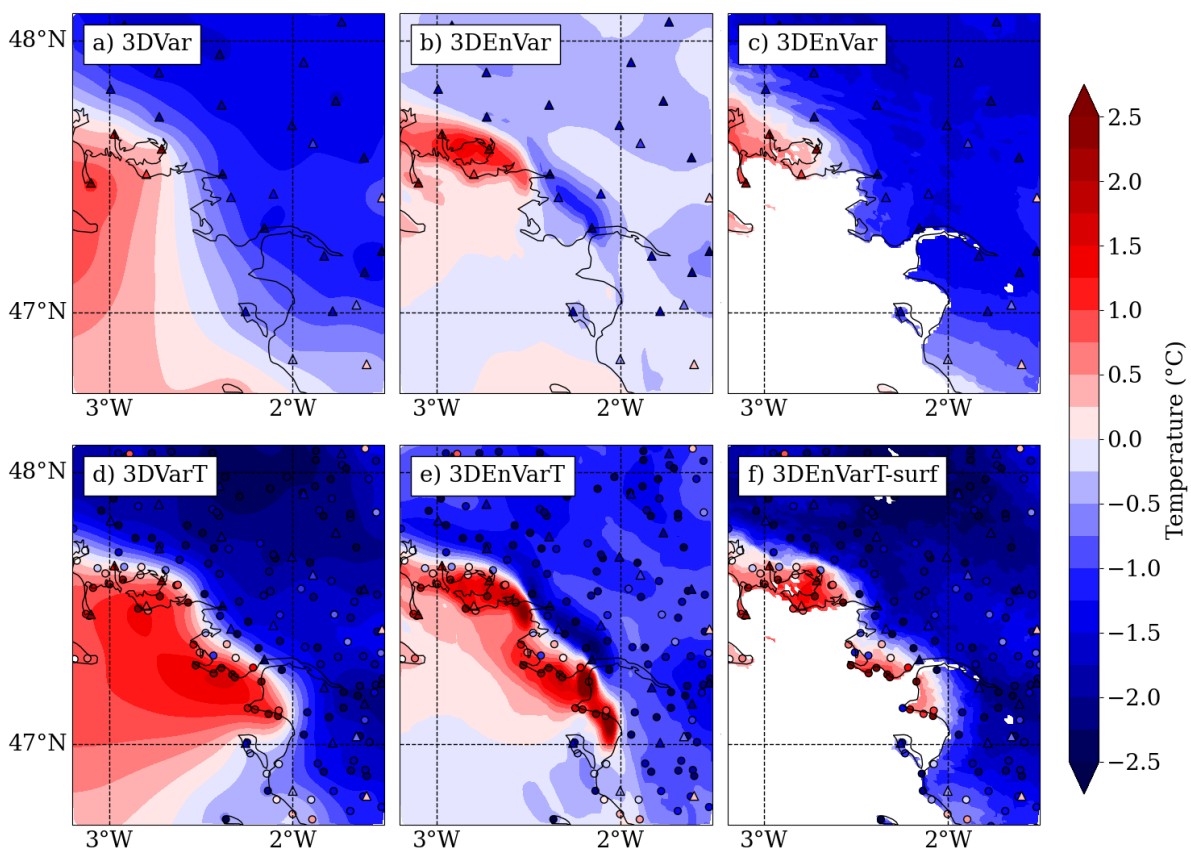

**Figure 11.** Temperature increments (a–b, d–e) at the lowest atmospheric model level (around 5 m height AGL) and (c, f) at the highest surface model level (ground at 0 m height AGL) on 7 September 2021 at 15:00 UTC. These experiments are designed as (a) 3DVar, (b, c) 3DEnVar, (d) 3DVarT, (e) 3DEnVarT and (f) 3DEnVarT-surf experiments, the only difference being that they all start from the same 3DVar background at 14:00 UTC. The location of stations whose observations of screen-level temperature are assimilated are indicated by triangles for SWSs and circles for PWSs; their colours indicate temperature OmBs.




temperature of these SWSs, respectively. The daily maximum temperature at Nantes-Bouguenais was 5 °C above or equal to the September normal from 2 to 8 September, i.e. for more than 5 consecutive days, which is the WMO criterion for heatwaves.

Figure 10 shows the temperature analysis at around 5 m height AGL, for the different experiments, and both SWS and PWS
screen-level temperature observations. If the height between observations and this analysis field is different, it is shown here because it is the lowest atmospheric level to which DA increments are applied. The experiments are cycled since 6 September 2021 at 00:00 UTC, which accounts for the significant discrepancy observed between the analysis. The two types of pre-processing show similar results for this case (not shown).

In the afternoon, a sea breeze rises up due to the difference in temperature between the ocean and the land, cooling the coast.
This cooling is observed by three SWSs, and well observed with PWSs, particularly after pre-processing (Figs. 9a and 10). Analyses from the four DA experiments (Figure 10) are able to reproduce the coastal cooling caused by the breeze, but with differences in temperature on either side of the breeze front and differences in location of this front.

In the 3DVar experiment (Fig. 10a) the temperature analysed is very close to SWS observations of temperature. When PWS temperature observations are assimilated in 3DVarT (Fig. 10c), the temperature analysed in the area A is lower than in 3DVar,
which increases the deviation from the analysis of the Vannes-Séné SWS observation. In the 3DEnVar experiment (Fig. 10b), still in the area A near Vannes-Séné SWS, the breeze front is located further inland which is not in agreement with this SWS observation or PWS observations shown in Fig 10c. Also, in the area B, temperature analysed in 3DEnVar is higher than SWS temperature observations. In this case, 3DEnVarT analysis seems closer than the other analyses to both PWS and SWS observations, in particular in areas A and B (Fig. 10d).

To illustrate the shape of temperature increments given by each DA scheme depending on the observations assimilated (from SWSs only or from both SWSs and PWSs), at the lowest atmospheric level for atmospheric DA schemes and at ground level for surface schemes, experiments starting from the same 3DVar operational background at 14:00 UTC are shown in Fig. 11.

Increments at the lowest atmospheric model level have lower horizontal gradients in 3DVar (Fig. 11a) w.r.t. 3DEnVar (Fig. 11b) and in 3DVarT (Fig. 11d) w.r.t. 3DEnVarT (Fig. 11e). Because the information from observations propagate at
longer distance with the settings of the 3DVar DA scheme than the 3DEnVar DA scheme (as shown in Sect. 2.1.1), increments of the same sign as the OmBs of coastal observations propagate over longer distances at sea in 3DVar and 3DVarT than in 3DEnVar or 3DEnVarT experiments.

The assimilation of PWS observations in 3DVarT and 3DEnVarT modifies the shape and sign of the increments when compared to 3DVar and 3DEnVar, in particular, extending positive increments southwards.

In the area B and generally inland, increments are much lower than the neighbouring SWS OmBs in 3DEnVar whereas it is not the case in 3DVar. As shown in Sect. 4, the 3DEnVar DA scheme gives less relative weight to surface temperature observations, which explains why observations have little impact on the temperature analysis. Assimilating PWS observations (Fig. 11e) increases the amplitude of increments, in particular inland, in 3DVarT w.r.t. 3DVar or in 3DEnVarT w.r.t. 3DEnVar.

At ground level, the shape of the ground temperature increments with the OI in 3DEnVar experiment (Fig. 11c) is similar to
the shape of the atmospheric increments in the 3DVar experiment. When assimilating PWS temperature observations with the



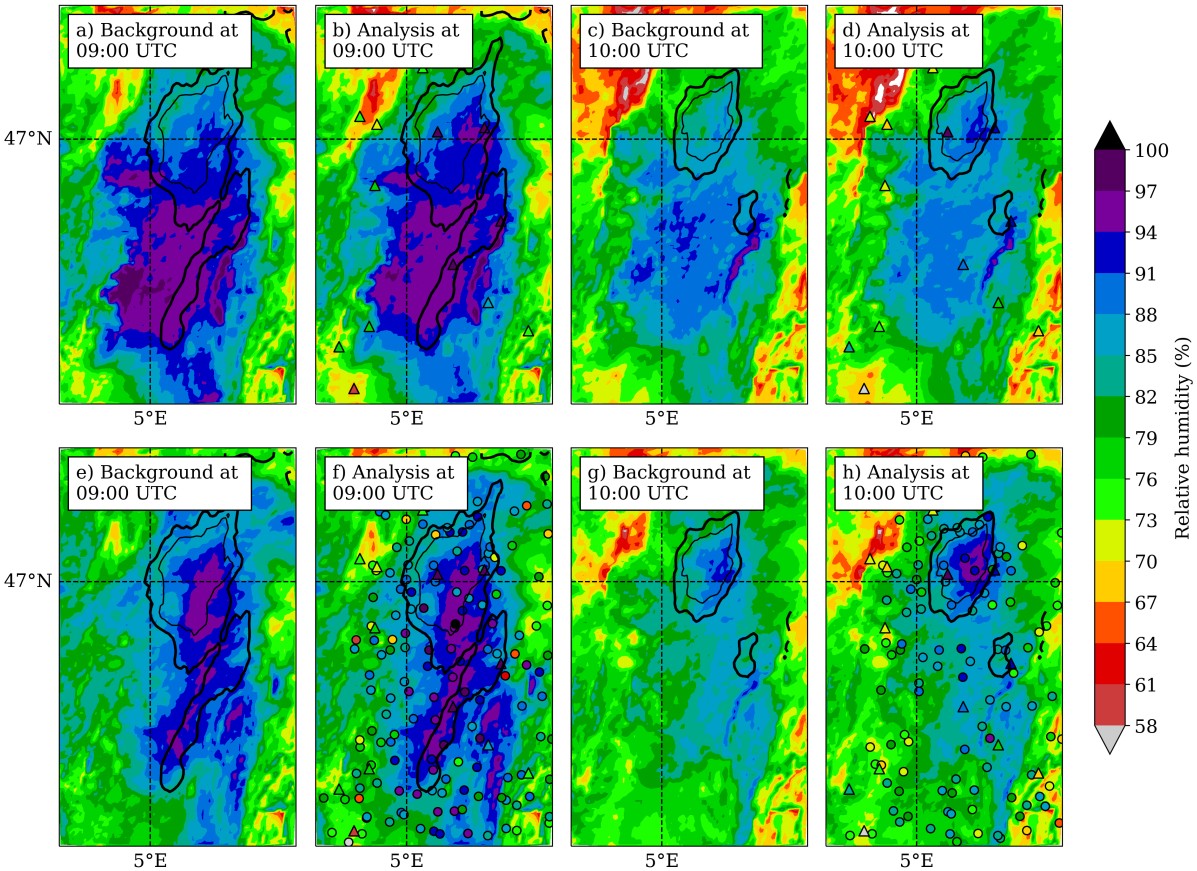

**Figure 12.** Relative humidity at the lowest atmospheric model level (around 5 m height AGL) for (a–d) 3DEnVar, and (e–h) 3DEnVarRHS experiments on 23 September 2021. (a, c, e, g) Backgrounds and (b, d, e, f) analyses are shown at two consecutive assimilation times: (a, b, e, f) 09:00 UTC and (b, d, f, h) 10:00 UTC. Assimilated observations of relative humidity from (triangles) SWSs and (circles) PWSs are indicated. MSG-3 satellite HRV channel reflectance observations equal to 24 % and 43 % are shown by bold and thin black edges, respectively.

OI, the sign of increments at the surface is generally the same as the sign of increments added in the atmosphere in 3DVarT or 3DEnVarT, and their values are closer to 3DVarT than to 3DEnVarT.

## 5.2 Fog on 23 September 2021

The Saône valley, in the northwest of France (Fig. 2), is regularly affected by fog during the autumn and winter seasons. Situated between the Morvan and the Jura mountainous massif, it is crossed by a few rivers, including the Saône (Fig. 9b), providing humidity near the surface. On 23 September 2021, high-pressure conditions are associated with stable air masses over France (not shown). At 09:00 UTC (Fig. 9b), MSG-3 satellite HRV observations indicate the presence of low clouds.




These clouds are reaching the ground, causing mist and fog: SWS and PWS relative humidity observations reach 90 to 100 %, three crowdsourced human observations indicate fog between 08:00 and 09:00 UTC and a forward scatter sensor at Tavaux

SWS, south-west of Dole, observes a meteorological optical range of 1020 m at 09:00 UTC, between mist and fog. Out of the fog, between 4.5 and 5° E, relative humidity observations range from 60 to 90 %.

Figure 12 shows the evolution of relative humidity in the backgrounds and analyses of two experiments at two assimilation times: 09:00 and 10:00 UTC.

At 09:00 UTC, at the lowest atmospheric level, relative humidity backgrounds of 3DEnVar (Fig. 12a) and 3DEnVarRHS

(Fig. 12e) experiments are different, due to the cycled assimilation. High values of relative humidity simulated by 3DEnVarRHS appear closer to the MSG-3 observation of clouds, in its east-west extension, than 3DEnVar. The analysis at 09:00 UTC (Fig. 12b, f) slightly increases the relative humidity under the observed fog, but is not able to substantially modify the shape of the wet area in the background in only one assimilation time step, in particular in 3DEnVar where only SWS observations are assimilated.

At 10:00 UTC, the 1 h forecast drastically reduces the wet area for both experiments (Fig. 12c, g). Whereas in 3DEnVar, the area with relative humidity above 91 % is still large in its east-west extension, in 3DEnVarRHS the wettest area is located where clouds are still observed by satellite. The assimilation at 10:00 UTC increases the relative humidity in this area, for both experiments, which exceeds 94 %, as indicated by one SWS, only in 3DEnVarRHS.

This example shows the importance of rapid-update cycles in DA, as one assimilation may have a low impact, but several are

gradually changing the simulated relative humidity. The model plays a preponderant role in reducing the simulated fog from 09:00 to 10:00 UTC. Here this situation is illustrated with 3DEnVarRHS because this experiment is closer to the observations than 3DEnVarRH. Regarding the results of 3DEnVarRH-surf, in this case, the assimilation of PWS observations in the OI scheme at the surface does not change the shape and the intensity of the simulated fog (not shown).

## 6   Discussion and conclusion

This study has explored the impact of the assimilation of temperature and relative humidity observations from PWSs in the current operational AROME system. Before the traditional screening stage of a DA system, two pre-processing methods of PWS observations were designed and selected. This was done to obtain pre-processed PWS observations with statistics of differences to the AROME background close to SWS observations currently assimilated in the AROME DA system.

– Of the four bias correction methods evaluated, two have been selected, including the method of Sgoff et al. (2022) de-
signed to reduce the diurnal bias of PWSs. Because these methods are based on statistics of differences between AROME background and observations, there is a risk of transferring model bias to PWS observations, hence the importance of assimilating anchor near-surface observations.

– Two quality control methods have been evaluated. The Titan-QC original design with three checks has been found detrimental when multiple PWS exhibit biases with respect to SWS observations; however, all checks described by Båserud



440       et al. (2020) such as the First guess test have not been tested. A QC adapted from Mandement and Caumont (2020) has been used to remove PWS observations judged erroneous based on their departures with the AROME background.

- A thinning has been applied to reduce the probability of having correlated observation errors, by taking one random observation per mesh in an approximately $8\,\mathrm{km}$ horizontally spaced regular Gaussian grid.

Then, the OSE framework has been used: the added value of PWS observation assimilation with two selected pre-processing

methods and existing DA schemes has been quantified. The assimilation of pre-processed PWS temperature observations with the 3DEnVar DA scheme reduced between $0.7$ and $0.9\,\%$ the rms SWS temperature OmBs in France, depending on the pre-processing method. Departures between SWS observations and forecasts are reduced up to 3 to $6\,\mathrm{h}$ range. Results are similar for PWS relative humidity observations, with a reduction of $1.2$ to $1.5\,\%$ of the rms SWS relative humidity OmBs in France. These findings are in agreement with benefits found by Sgoff et al. (2022), in experiments where PWS and SWS observations

were not jointly assimilated.

Finally, the assimilation of pre-processed PWS observations with the 3DEnVar DA scheme has been subjectively found to better represent the fine-scale features of a sea breeze during a heatwave event, and the evolution of a fog event. In these case studies, the relative weight given to near-surface observations is found to be lower in 3DEnVar compared to 3DVar, which is compensated by the high amount of PWS observations assimilated.

Two distinct bias corrections have been evaluated. The monthly bias correction (BC-M) demonstrates superior performance in reducing the departure between SWS observations and the model background. In contrast, the two-day temporal window bias correction (BC-S) exhibits adaptive correction capabilities that have been shown to be relevant in case studies. Further tests on different seasons would be beneficial in order to validate the results.

This study has some limitations. The OmBs used for P-M and P-S pre-processing for all experiments are computed from a

monitoring experiment with a 3DVar DA scheme (Monitor). A more rigorous calculation would require a monitoring experiment to be carried out for each assimilation scheme tested. To determine the thinning distance, the spatial Desroziers diagnostic (Fig. 7) is computed from an experiment with a 3DEnVar DA scheme. This thinning distance is applied to the experiments with a 3DVar DA scheme as well. Perhaps the scores of the 3DVar experiments are degraded by an inappropriate choice of the thinning length, which could be increased in future studies. Furthermore, no testing has been specifically conducted in

complex topographical areas, such as mountainous regions, where the pre-processing may deteriorate the new observations or reject them due to temperature model bias (Gouttevin et al., 2023).

The joint assimilation of temperature, relative humidity, and surface pressure observations from PWSs in AROME, could now be tested, as these variables describe weather structures more coherently together than separately. Efforts should be made to evaluate such joint assimilation over a wider variety of cases and perhaps over longer study periods.

For operational use, beyond the recommendations of D24 which also apply in this case, this study has not attempted to modify the settings of the assimilation systems used (3DVar, 3DEnVar or OI), and the few new settings have been chosen identically to the operational ones. However, studies of the sensitivity of these settings would be desirable, given their importance in the results obtained and because they are set to obtain the best scores with the observations currently assimilated. In the atmo-





sphere, while the 3DEnVar system is currently only being tested, there are plans to replace it rapidly with a 4DEnVar system:
it would allow more observations to be included in the analysis by making it possible to assimilate time series of observations.
At the surface, the replacement of the OI by a 2DEnVar system is currently under study. This would allow more observations
to be assimilated at the surface and ensure greater consistency between atmospheric and surface schemes: it is also a first step
before possibly coupling surface and atmospheric DA systems.

## Appendix A: Relative humidity computation

The relative humidity ($RH$) is calculated in AROME as:

$$RH = \frac{100Pq}{e_{\text{sat}}(\frac{R_{\text{dry}}}{R_{\text{vap}}} + q(1 - \frac{R_{\text{dry}}}{R_{\text{vap}}}))} \tag{A1}$$

$P$ is the pressure in Pa, $q$ is the specific humidity (dimensionless), $R_{\text{dry}} = 287.0597 \, \text{J} \, \text{kg}^{-1} \, \text{K}^{-1}$ and $R_{\text{vap}} = 461.5250 \, \text{J} \, \text{kg}^{-1} \, \text{K}^{-1}$
are the gas constants for dry air and water vapour, respectively.

The saturation vapour pressure $e_{\text{sat}}$ in Pa is equal to:

$$e_{\text{sat}} = \exp\left(\alpha_l + (\alpha_s - \alpha_l)\delta - \frac{\beta_l + (\beta_s - \beta_l)\delta}{T} - (\gamma_l + (\gamma_s - \gamma_l)\delta)\ln(T)\right) \tag{A2}$$

with $\delta = 0$ if $T \geq 273.16 \, \text{K}$ and $\delta = 1$ if $T < 273.16 \, \text{K}$ and,

$$\alpha_l = \ln(e_{\text{sat}}(T_t)) + \frac{\beta_l}{T_t} + \gamma_l \ln(T_t) \qquad \beta_l = \frac{L_v(T_t)}{R_v} + \gamma_l T_t \qquad \gamma_l = \frac{C_l - C_{\text{pv}}}{R_v} \tag{A3}$$

$$\alpha_s = \ln(e_{\text{sat}}(T_t)) + \frac{\beta_s}{T_t} + \gamma_s \ln(T_t) \qquad \beta_s = \frac{L_s(T_t)}{R_v} + \gamma_s T_t \qquad \gamma_s = \frac{C_s - C_{\text{pv}}}{R_v} \tag{A4}$$

$T$ is the temperature in K, $T_t = 273.16 \, \text{K}$ is the triple point temperature, $L_v(T_t) = 2.5008 \times 10^6 \, \text{J} \, \text{kg}^{-1}$ is the specific
latent heat of vaporization, $L_s(T_t) = 2.8345 \times 10^6 \, \text{J} \, \text{kg}^{-1}$ is the specific latent heat of sublimation, $e_{\text{sat}}(T_t) = 611.14 \, \text{Pa}$,
$C_l = 4218 \, \text{J} \, \text{kg}^{-1} \, \text{K}^{-1}$ is the specific heat for water in its liquid phase, $C_s = 2106 \, \text{J} \, \text{kg}^{-1} \, \text{K}^{-1}$ is the specific heat for water in
its solid phase and $C_{pv} = 4R_{\text{vap}}$ is the specific heat at constant pressure and constant volume for water vapour.

## Appendix B: Comparison of observations and AROME background by triple collocation

PWS and SWS screen-level observations are considered collocated when separated by less than $1 \, \text{km}$ horizontally, $50 \, \text{m}$ vertically and less than $10 \, \text{min}$ apart in time. When it happens, the AROME background from the Monitor experiment (Table 1) corresponding to this SWS observation, given by the observation operator, provides a third value of the physical variable considered ($T_{2\text{m}}$ or $RH_{2\text{m}}$). This triple collocation method (Stoffelen, 1998) is performed over the 1-month study period, giving a sample of 536000 (respectively 80000) triplets of collocated values for temperature (resp. relative humidity).

For each variable, three triplets with three different pre-processing methods for PWS observations are tested: (i) raw i.e. without pre-processing (ii) P-M without thinning and (iii) P-S without thinning. We will refer to them simply as P-M and P-S in this paragraph only. Figure B1 shows distributions of pairwise differences between the values in each triplet.





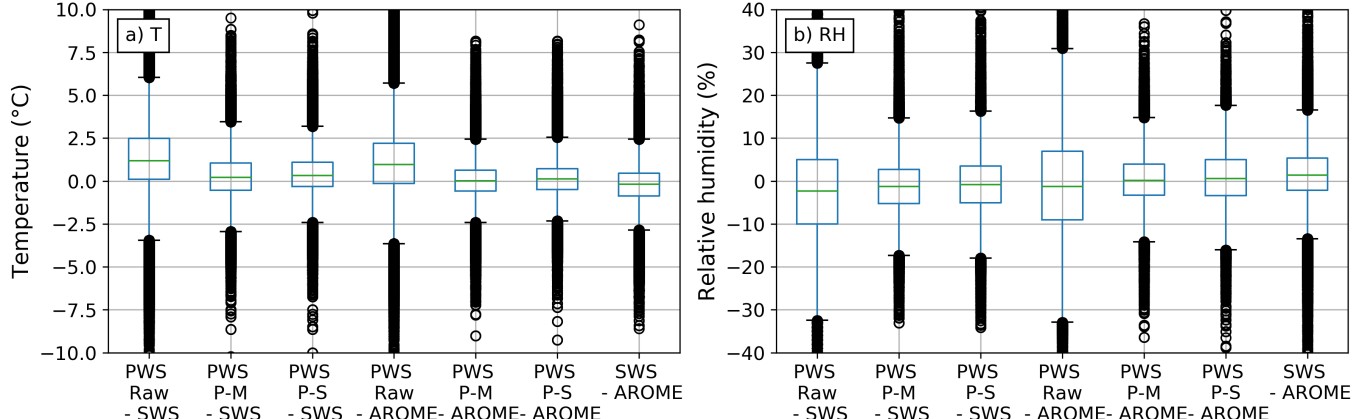

**Figure B1.** Boxplots showing distributions of pairwise differences of (a) $T_{2m}$ and (b) $RH_{2m}$ in triplets of collocated values (SWS observations, PWS observations, AROME background from Monitor), depending on the pre-processing (raw, P-M without thinning, P-S without thinning) applied to PWS observations.

Regarding the distributions of temperature differences between PWS and SWS collocated observations, their standard deviation is substantially reduced when pre-processing are applied, from 2.3 °C (raw) to 1.3 °C with P-M or 1.3 °C with P-S. Also, mean differences are closer to 0 and the interquartile range is reduced.

Regarding the distributions of temperature differences between the pre-processed PWS observations, with both P-M and P-S, and the AROME background, the standard deviation equals to 1.0 °C which is smaller than 1.3 °C, the standard deviation of differences between the pre-processed PWS observations and SWS observations. This could show that the selected pre-processing methods of PWS observations overfit the AROME background.

Similar conclusions can be drawn for relative humidity, although the effects of PWS pre-processing do not reduce interquar-
tile ranges of differences with SWS observations or the AROME background by the same factor as for temperature.

*Author contributions.* Alan Demortier: Formal investigations, methodology, analysis; resources; writing – original draft. Marc Mandement: Formal analysis; resources. Vivien Pourret: Formal analysis; software; validation; supervision. Olivier Caumont: validation; project administration; supervision; validation.

*Competing interests.* The authors declare that they have no conflict of interest.

*Code availability.* The code used for the assimilation experiments in AROME-France is owned by the members of the ACCORD consortium. This agreement allows each member of the consortium to license the shared ACCORD codes to academic institutions in their home countries



for non-commercial research. Access to ACCORD codes, and codes used for the figures, can be obtained by contacting the corresponding author.

*Data availability.* WMO essential weather station observations (SYNOP and SHIP reports), French RADOME observation reports, AROME
analyses and forecasts are available freely online (Météo-France, 2024a, b, c, d). PWS observations are provided freely by Infoclimat (2024) for the StatIC network and Toulouse Métropole (2024) for the Toulouse Métropole network; freely in real-time, and archives on request by Netatmo (2024) for the Netatmo network and they cannot be shared according to their licence: https://dev.netatmo.com/legal. Satellite MSG-3 HRV observations used for this study (corrected by Météo-France) are available on request from the corresponding author, level 1.5 observations are provided by EUMETSAT (2024). The results of data assimilation experiments are available on request from the
corresponding author solely for non-commercial research purposes and for reasonable data volumes.

*Acknowledgements.* The authors would like to thank Jean-Baptiste Hernandez from the Centre de Météorologie Spatiale in Lannion for information provided on satellite observations.

*Financial support.* This work was supported by the French National program Les Enveloppes Fluides et l'Environnement (LEFE), projects ASMA and EXDOMO.



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
