# Peer review of "Assimilation of temperature and relative humidity observations from personal weather stations in AROME-France"

_EGUsphere, 2024_

## Author Response (AR1)

**Reply to referees - nhess-2024-1673 - "Assimilation of temperature and relative humidity observations from personal weather stations in AROME-France"**

Parts added to the manuscript are in blue and deleted parts are in . Line numbers (L) in our replies are those of the preprint.

**Reply to anonymous referee #1**

**General comments**

*The manuscript "Assimilation of temperature and relative humidity observations from personal weather stations in AROME-France" analyses the value of assimilating T2M and RH2M observations of PWS in the AROME-France NWP model. To answer the questions of how the data should be pre-processed and what influence the DA scheme has, assimilation cycles of one month and two case studies were evaluated. In the case studies, the impact of assimilation on the prediction of a heat wave and a fog event was analysed. The procedure presented in the manuscript is reasonable and well structured. Therefore I only have some minor questions.*

> We would like to thank the first anonymous reviewer for his review and feedback.

**Specific comments**

*1. I miss the information which observation types are assimilated besides the observations of the PWS. It would be great if you could mention which observations are assimilated in your experiments 3DVar and 3DEnVar.*

> Indeed, the other assimilated observation types are not mentioned. We have added (L91): In the AROME atmospheric DA system, an average of 49253 observations (active, see below) are assimilated per hour during the one-month study period. A total of five observation systems account for 99 % of incoming active observations: radar (56.5 %), SWSs (17.0 %), satellite (9.8 %), radiosoundings (9.6 %), and aircraft (6.1 %). Among the variables observed by SWSs, the ones assimilated in AROME are screen-level temperature and relative humidity, zonal and meridional wind at 10 m height AGL and the surface geopotential. More details are given in Sect. 2.2.1.

*2. Why do you decide to do experiments with T2M/RH2M only and no experiment with combined assimilation of RH2M and T2M PWS stations?*

> The study presents the assimilation of both T2M/RH2M observations separately, as a continuation of the previous article on pressure observations. The impact on the forecasts is very different for each parameter. It is necessary to understand each of them well before assimilating them together, in order to be able to differentiate their impact. The joint assimilation of the three types of observations is considered a separate scientific question, which should be addressed in future work, as mentioned L467–L469.

*3. Line 399: For me it is unclear if Fig 11c is with or without OI. The sentence: At ground level, the shape of the ground temperature increments with the OI in 3DEnVar experiment (Fig. 11c) is similar to ... suggests that Fig 11c is with OI, but in the Figure caption it reads as it is 3DEnVar only.*

> Figure 11c shows surface (0 m height AGL) temperature increments calculated by the OI DA scheme, of the experiment named 3DEnVar (Table 1). Experiments are named only from their altitude DA scheme which may cause confusion, given that there is no column about the surface DA scheme in Table 1. To clarify, we have modified the caption of the Table 1 describing the experiments: All experiments use the OI surface DA scheme. *X* corresponds to the variable of PWS observations assimilated in the atmospheric DA scheme only unless otherwise indicated: T for $T_{2m}$ or RH for $RH_{2m}$.

**Technical corrections**

*Line77f: Sometimes Section and sometimes Sect.*

> The abbreviation has been maintained according to the publisher's guidelines (*Copernicus Publications*, 2023).

40      *Figure2: A better contrast between observations and the domains would be great. It is difficult to read the numbers of the domains.*

> You are right. We have increased the contrast.

     *Figure4a: y-axis should be labeled with Mean OmB (°C)*

> We have modified it.

45      *Figure6: It is not mentioned what is represented by the coloured stars.*

> We have modified it:  Coloured squares, stars and triangles are Toulouse Métropole PWS, StatIC PWS and SWS screen-level temperature observation, respectively.

     *Figure10: The meaning of circle A and B is missing in the Figure caption.*

> We have added : Encircled areas A and B are referred to in the text.

50      *Figure11: Column titles of the shown temperature level would be useful. Label of colourbar should be temperature increments.*

> The labels of Fig. 11 have been modified as follows to include the shown temperature level: (a) 3DVar (5 m) (b) 3DEnVar (5 m) (c) 3DEnVar (0 m) (d) 3DVarT (5 m) (e) 3DEnVarT (5 m) (f) 3DEnVarT-surf (0 m). We have modified the colour bar according to the recommendations. We thank the reviewer for the advice.

**Reply to anonymous referee #2**

     *The authors have performed detailed experiments of data assimilation of PWS data into the AROME model, testing various DA systems and bias corrections of the PWS data to improve forecasts under certain circumstances. I personally enjoyed reading this paper: it's very thorough in its methodology and is critical of the limitations that DA systems and PWS ingestion into models have whilst accounting for those limitations and trying to solve inherent biases and issues.*

> We would like to thank the second anonymous reviewer for his review and his positive feedback.

     *While the paper is methodologically and scientifically sound, I do struggle to read through it at times, especially at the start and in the methodology section, since there is a very widespread use of jargon and abbreviations, most of which are not used often enough to justify their use. The high amount of abbreviations and numbers being thrown around in the text makes it a challenging read and risks losing the main message in a difficult-to-read section, which would be a shame. Please reconsider the amount of abbreviations, especially for the non-intuitive ones such as OmB. Also some of them are written out multiple times along the text. My advice would be to only keep those that come back constantly (PWS) or are essentially common in usage (QC). This will make the text much more readable.*

70     > We agree. Outside the abstract, multiple terms written out in full have been removed for the following abbreviations: PWS, SWS, QC-MC or QC-Titan. Abbreviations that are not used enough to justify their use have also been removed:  (L25, L341) ,  (L46) ,  (L61)  (L117),  (L146, L148),  (L151),  (L195), ,  (L226, L228). However, we wish to keep abbreviations such as OmB because "observation minus background" is very long and to be consistent with D24.

75     *Other than that I think the paper is quite good already, I have listed some more significant comments below that came to mind during my reading. Some of which are more questions borne out of curiosity, and some are points I had some issue with understanding properly. If the writing style is cleaned up and the issues below are addressed, I think the paper is well worth of publishing.*

**Specific comments**

80     *Squall lines are mentioned, but rainfall observations are not included in the DA? Is this by choice or because the DA system for rainfall typically uses reflectivity and not on-ground data? I think it would be good to reflect on this since there has been substantial work on the quality of PWS rainfall data as well, especially since rainfall data is quite strongly spatially distributed.*

    > Rainfall observations from SWS rain gauges are not operationally assimilated in AROME-France, even if a method to
85 assimilate a precipitation analysis merging radar and rain gauge data was developed by *Sahlaoui et al.* (2020). They are only used for model verification. As hydrometeors are planned to become a control variable in AROME-France, the improvement of current methods and the joint assimilation of SWS and PWS rain gauge observations could become in the future the subject of work that is beyond the scope of this study. We have added (L479) : Finally, further work remains to determine the potential for assimilating observations from PWS anemometers and rain gauges, with specific pre-processing.

90     *Most PWS stations are urban – how is that dealt with in the DA schemes, given urban roughness parameters and so on? Since these areas are not WMO compliant as would be the case for standard DA of observations.*

    > In this study, PWS observations are assimilated exactly as SWS ones, so this issue is not treated differently. This seems to us to be a good idea, because not all SWS observations currently assimilated have the same level of compliance with WMO standards. We've added this sentence L101 to explain it: Thus, the effect of the type of surface on observations is taken into
95 account by these observation operators, either directly when it includes a surface variable modelled by SURFEX (for $T_{2m}$), or indirectly when it only includes atmospheric variables, modified through the surface-atmosphere exchanges during the forecast (for $RH_{2m}$).

    *RH as [diagnostic] variable calculated whereas it is directly measured by PWS (with inherent autocorrelation to the Tair measurements). How does that influence DA procedure? Also, RH depends on temperature, meaning*
100     *biases are autocorrelated between the two, I didn't see that point discussed (or perhaps I've missed it).*

    > We agree that $T$ and $RH$ PWS observation errors can be correlated. This also concerns SWSs. In the DA procedure, when $T_{2m}$ observations are rejected by the screening, then $RH_{2m}$ observations are also rejected. We have added (L469): In these joint experiments, the correlation of observation errors, especially between $T_{2m}$ and $RH_{2m}$ observations, should be examined. Correlated observation error issues could be addressed by modelling a non-diagonal observation error covariance matrix or by
105 inflating observation errors.

    *Where does the sigma-o level in the OmB calculation come from? What is its influence on the bias correction successes? In L. 110: I don't quite understand how the rejection threshold is calculated in the given examples – what does it mean that the threshold is "up to 6.5 °C"? Is the threshold something different than the OmB?*

    > In the DA procedure, the observations that are too far from the background (i.e. one-hour model forecast) are rejected
110 according to Eq. 1. The rejection threshold is calculated with the use of $\sigma_o$ , which is prescribed, and $\sigma_b$ which is calculated.

This threshold is thus fluctuating, depending on $\sigma_b$ , and for the first hour of experiment is going up to 6.5 °C. In other words, for this hour, observations that are more than 6.5 °C away from the background are rejected (i.e. $OmBs > 6.5$ °C). We have added (L108): the right-hand term is referred to as the rejection threshold

*154: the number of observations is given, but does that mean that the amount of stations is equal to that? Or do stations give multiple obs. per hour?*

> The number of stations is equal to the mentioned number of observations. Even if, stations may give observations up to 5 min frequency, only one observation is assimilated per hour. We have modified it (L154) :  In average over one month, the number of SWSs providing at least one observation per hour of temperature and relative humidity is 2440 and 1600, respectively.

*P 6/7: do you also have an indication on the amount of assimilated Netatmo stations, as you do for the other networks? Would put things into perspective and point out the importance of these data, as these numbers can be huge from my own experience. I see figure 3 contains frequency diagrams but a (rough estimation of) number of stations would be helpful.*

> Indeed, we do not explicitly mention the number of Netatmo stations, except on labels in Fig. 3. We have added (L164): In average over one month, the number of PWSs providing at least one observation per hour of temperature and relative humidity is 63099 and 63117, respectively.

*The assumption that observational errors are not correlated is an interesting one, as there can be apparent errors caused by urban morphology, i.e. microclimatical variations caused by the urban environment that are only picked up by a set of sensors at the specific location, whereas all neighbours would not find it, therefore classifying it as an error. How robust is the QC system that it does not dismiss microclimatical variations as sources of error? And how important is this for the DA procedure?*

> We have developed this limitation in Sect. 6 (L466): In DA, the observation error includes two contributions: a measurement error and a representativity error. The assumption that observational errors are uncorrelated becomes less and less valid as the average spacing between observations decreases, requiring the use of thinning techniques. Microclimatic variations, for example in urban environments, are currently considered as representativity errors if we cannot model them, and are therefore also considered to be representativity errors by QC algorithms based on OmB thresholds. Advances in surface modelling, the refinement of horizontal grids and work on modelling the correlation of observation errors should gradually reduce the proportion of microclimatic variations considered as a source of error.

*Section 4.1: this could benefit from a spatial image as well to complement figure 8, since I'm curious what the effect of PWS density on forecast quality is.*

> We agree. Figure A shows the $\Delta$rms OmB (%) of $T_{2m}$ of 3DEnVarTS w.r.t. 3DEnVar. The absolute value of $\Delta$rms OmB is higher over France, where PWS observations are assimilated, but there is no clear geographical structure even if improvement in blue dominates. Areas with the largest PWS density do not clearly show more improvement than elsewhere, probably given the thinning applied at 8 km scale. Not shown as we rather prefer not making the article any longer.

*Figure 11: It's not quite clear to me what is meant with 'temperature increment' (which is also not correctly labelled on the colour bar, as a side note).*

> We have modified the figure and corrected the colour bar label in response to this comment and that of the first reviewer. In DA, the increment is the difference between the analysis and the background. 'Temperature increments' is defined L117 and the notion is illustrated in Fig. 1.

[Figure]

**Figure A.** $\Delta$rms OmB (%) of $T_{2m}$ of 3DEnVarTS w.r.t. 3DEnVar during the one-month study period. Negative (resp. positive) values indicate improvement (resp. degradation).

**References**

Copernicus Publications, Natural Hazards and Earth System Sciences–Submissions, https://www.natural-hazards-and-earth-system-sciences.net/submission.html (last access: 7 November 2023), 2023.

Sahlaoui, Z., S. Mordane, E. Wattrelot, and J.-F. Mahfouf, Improving heavy rainfall forecasts by assimilating surface precipitation in the convective scale model AROME: A case study of the Mediterranean event of November 4, 2017, *Meteorological Applications*, *27*(1), e1860, https://doi.org/https://doi.org/10.1002/met.1860, 2020.